# RETRACTED: Curative Effect of Catechin Isolated from *Elaeagnus Umbellata* Thunb. Berries for Diabetes and Related Complications in Streptozotocin-Induced Diabetic Rats Model

**DOI:** 10.3390/molecules26010137

**Published:** 2020-12-30

**Authors:** Nausheen Nazir, Muhammad Zahoor, Riaz Ullah, Essan Ezzeldin, Gamal A. E. Mostafa

**Affiliations:** 1Department of Biochemistry, University of Malakand, Chakdara Dir (L), Khyber Pakhtunkhwa 18800, Pakistan; mohammadzahoorus@yahoo.com; 2Department of Pharmacognosy, College of Pharmacy, King Saud University, Riyadh 11451, Saudi Arabia; rullah@ksu.edu.sa; 3Department of Pharmaceutical Chemistry, College of Pharmacy, King Saud University, Riyadh 11451, Saudi Arabia; esali@ksu.edu.sa (E.E.); gmostafa@ksu.edu.sa (G.A.E.M.); 4Micro-Analytical Laboratory, Applied Organic Chemistry Department, National Research Center, Dokki, Cairo 12622, Egypt

**Keywords:** *Elaeagnus umbellata*, catechin, streptozotocin, hyperglycemia, hyperlipidemia, antioxidant enzymes

## Abstract

In this study, catechin (CTN) isolated from *Elaeagnus umbellata* was evaluated for in vitro antioxidant potential and inhibition of carbohydrate digestive enzymes (α-amylase and α-glucosidase). The compound was also tested for its in vivo antidiabetic potential using Sprague-Dawley rats as experimental animals. The effects of various doses of catechin in STZ (Streptozotocin) induced diabetic rats on fasting blood glucose level, body weight, lipid parameters, hepatic enzymes, and renal functions were evaluated using the reported protocols. The CTN exhibited the highest percent antioxidant for free radical scavenging activity against DPPH and ABTS free radicals, and inhibited the activity of carbohydrate digestive enzymes (with percent inhibition values: 79 ± 1.5% α-amylase and 80 ± 1.1% α-glucosidase). Administration CTN and standard glibenclamide significantly decreased the fasting blood glucose level and increased the body weight in STZ-induced diabetic rats. CTN significantly decreased the different lipid parameters, hepatic, and renal function enzyme levels along with Hb1c level in diabetic rats, while significantly increasing the high-density lipoprotein (HDL) level with values comparable to the standard glibenclamide. Further, the altered levels of glutathione and lipid peroxides of liver and kidney tissues were restored (by CTN) to levels similar to the control group. CTN significantly increased the antioxidant enzyme activities, total content of reduced glutathione, and reduced the malondialdehyde (MDA) level in rat liver and kidney tissues homogenates, and also corrected the histopathological abnormalities, suggesting its antioxidant potential.

## 1. Introduction

Type 2 diabetes mellitus (T2DM) is a metabolic disorder that increases glucose level in blood which ultimately reduces life expectancy, diminishes life quality, and finally leads to mortality and morbidity [1]. Around 25% of the world’s population in the developing and developed countries are suffering from diabetes mellitus. To decrease the burden of glucose and related complications in diabetes, glycemic control is important from a clinical point of view [2]. DM is considered as a major chronic disease after cancer and cardiovascular diseases in human, caused either due to insufficient insulin secretion by pancreatic islet cells of Langerhans or due to insulin resistance that leads to hyperglycemia [3]. Increase in blood glucose is associated with the activities of two intestinal enzymes; α-amylase and α-glucosidase that causes the degradation of carbohydrates into disaccharides and finally to monosaccharide. Plasma glucose can be regulated by inhibiting these two enzymes [4] as DM is associated with hyperlipidemia due to impairments in metabolic pathways [5].

Reactive oxygen species (ROS) oxidizes the biologically important macromolecules, such as proteins, lipids, and nucleic acids, thus initiating structural and functional alterations in these molecules. Lipid oxidation produces a secondary product called malondialdehyde (MDA). Lipid peroxidation is a valuable target in the evaluation of oxidative stress and related complications because the hydroxyl radical produced is the most reactive form of ROS that can initiate lipid peroxidation by attacking polyunsaturated fatty acids. Oxidative stress is the major cause of diabetes mellitus and associated comorbidities [6].

Glycated haemoglobin (HbA1c) is a useful test that plays a central role in the diagnosis, prevention, and monitoring of diabetes. The benefit of glycated haemoglobin testing is that it does not vary noticeably with time and can therefore be done at any time of the day and also provides better reproducibility than glucose testing. HbAc1 testing provides instant results that are very important for therapeutic purpose and also for lifestyle interventions and improve overall diabetes care and control [7]. The American Diabetes Federation uses an HbA1c value greater than 6.5% as a definite diagnosis of diabetes mellitus [8]. Furthermore, HbA1c levels between 5.7% and 6.4% correspond to people with pre-diabetes who are at higher risk of developing DM.

According to the world health organization (WHO) evaluation report, medicinal plants are of great importance as they are sources of safer modern and effective drugs, with comparatively low side effects. Furthermore, pharmacological and chemical investigations are necessary to explore potential antidiabetic agents from plants sources [9]. The ideal aim to control T2DM is an effective control of blood glucose level, HbA1c, hypertension, and hyperlipidaemia [10].

Due to the scientific exploration on berries in the recent years they have gained more importance for human beings, as they have rich phytoconstituents and have great health benefits. The key flavonoids found in berries are proanthocyanins, anthocyanin, catechins, and flavonols, while the hydroxylated phenolic acids are cinnamic and benzoic acid that can reduce hyperglycemia through regulation of α-glucosidase and α-amylase enzymes [11]. Many studies from the literature have revealed that regular consumption of flavonoid rich berries fruit can alleviate type 2 diabetes, ischemic diseases, aging effects, Parkinson’s, and Alzheimer’s disease [11]. Many authors have reported the beneficial effects of berries fruits, vegetables and fruits juices that have exhibited positive health benefits, recommending them to be a good source of phytochemicals that could be used for curing type 2 diabetes and associated hypertension [12]. To properly manage T2DM, further work is needed to isolate potent compounds from medicinal plants with low side effects.

Among the berried plants, *Elaeagnus umbellata* Thunb, reported as an antidiabetic medicinal plant [13], is a spiny deciduous shrub belonging to the family *Elaeagnaceae.* The plant is native to South Europe, Central Asia, Japan, China, India, Afghanistan, Azad Kashmir, and Himalayan areas of Pakistan [14,15]. Its berries fruit is a big source of antioxidant compounds (lycopene, β-carotene, lutein, phytofuluene, phytoene, β-cryptoxanthin, and α-cryptoxanthin), polyphenols, and vitamin E content (α-tocopherols and γ-tocopherols) with therapeutic significances [16,17]. Carotenoids and lycopene are the major sources for the astringent flavor of the berries fruits [18]. Recently, attention has been focused on the relationship between berries fruits and the inhibition of the DM progress.

Flavonoids are the most potent antioxidant identified in significant amounts in fruit and vegetable. Catechin is one of the most important flavonoids, with a comparatively high antioxidant activity [19]. Catechins have the beneficial effect of preventing skin damage. Catechins are important ingredients in the tea plant that have exhibited intensive anti-oxidant and other representative physiological activities in experimental animals. They are members of the group of polyphenol compounds found in many medicinal plants. The major sources of catechin are *Camellia sinensis* (*C. sinensis*) and *C. assumica.* Green tea contains 75–80% water and polyphenol (flavanols, flavandiols, flavonoid, and phenolic acid) compounds [19]. Green tea is a popular beverage consumed worldwide for its long history and numerous health benefits due to the presence of various amounts of catechins such as catechin, (–)-epigallocatechin-3-gallate, (–)-epicatechin-3-gallate, (–)-epigallocatechin, and (–)-epicatechin. Several investigational studies have indicated that the antioxidant properties of green tea come from the flavonoids. Catechins provide several health advantages by scavenging free radicals thus retarding extracellular matrix degradation induced by ROS, ultraviolet radiation and pollution. The health raising properties of catechins are commonly related to their redox antioxidant capacity rather than to the other molecular activities. Catechins are one of the most important flavonoids, with antioxidant and anti-inflammatory properties rendering them a potent therapeutic for diseases where the mentioned processes are critical negative elements [20]. A number of scientists have isolated catechin from different plants and evaluated them to be beneficial in controlling oxidative stress, diabetes, high blood pressure, lipid absorption and glucose intolerance induced by a high fat diet [19,21,22].

The present study was therefore planned to investigate the biological effects of *E. umbellata* berries catechins on blood glucose level, lipid profile, Hb1c level, hepatic and renal functions, antioxidant enzymes, and MDA levels in the STZ-induced diabetic rats.

## 2. Results

### 2.1. In Vitro Antioxidant Potential of CTN

DPPH inhibition potential of CTN were 65 ± 1.35***, 50 ± 1.1***, 44 ± 0.5***, 37 ± 0.4***, 31 ± 1.0***, and 29 ± 1.4*** at concentration 1000, 500, 250, 125, 62.5, and 31.25 µg/mL with their IC_50_ values 65µg/mL. While the ABTS free radical scavenging potential of CTN were 59 ± 1.5**, 46 ± 1.8***, 41 ± 0.8***, 36 ± 1.4***, 28 ± 1.2***, and 21 ± 1.5*** at concentration 1000, 500, 250, 125, 62.5, and 31.25 µg/mL with IC_50_ values 72µg/mL. The results indicated in Figure 1A,B and Appendix A indicate that CTN has comparable % inhibition potential of the used free radicals to standard ascorbic acid against (ascorbic acid IC_50_; 30 µg/mL against DPPH and IC_50_; 32 µg/mL against ABTS). The experiments have been performed in three replicates.

### 2.2. In Vitro Antidiabetic Potential of CTN

Highest percent inhibitory potential of diabetic enzyme (α-amylase; 83 ± 1.5, α-glucosidase; 85 ± 1.1) were exhibited by CTN at highest concentration 1000 µg/mL (Figure 1C,D; Appendix A). The IC_50_ values were calculated by evaluating the plot of % α-amylase and % α-glucosidase enzyme inhibition as a function of CTN concentrations. The IC_50_ values for α-amylase and α-glucosidase were 38 and 32 µg/mL respectively. These values are comparable to that of the standard acarbose was used as a positive control which causes percent inhibition 90 ± 0.4 and 89 ± 0.5 against α-amylase α-glucosidase at maximum concentration of 1000 µg/mL with IC_50_ value 30 and 26 µg/mL respectively (Appendix A).

### 2.3. In Vivo Antidiabetic Potential of CTN

#### 2.3.1. Effect of Various Treatments on Blood Glucose Level

Figure 2A,B depicts the effects CTN in various treatments groups and STZ group on fasting blood glucose level in rats. Treatment with STZ significantly increased fasting blood glucose level in rats of groups II (Diabetic control) on day 1st in comparison to the normal control (group I) group and CTN treated groups IV-IX. Blood glucose level of diabetic rats increased significantly (*p* < 0.001) on days 1, 5, 8, 10, 15, and 21 compared to normal control rats (groups I). In standard glibenclamide (Group III) treated group significantly (*p* < 0.001) decreased in fasting glucose level in blood of diabetic rats was observed (as compared to diabetic control rats on days 10, 15, and 21). Although CTN at lowest dose of 2, 5, and 10 mg/kg body weight exerted a significant effect on blood glucose level in diabetic rats when compared to diabetic control rats but the effect was more pronounced when bioactive compound was administered in high dose range (15, 30 and 50 mg/kg) and fasting blood glucose level of diabetic rats were significantly (*p* < 0.01, *p* < 0.001) decreased. Blood glucose level reduction was observable from the 5th day and onward while the patterns were significantly apparent on days 10 (*p* < 0.05), 15, and 21 (*p* < 0.001).

#### 2.3.2. Effect of Various Treatments on Body Weight in Diabetic Rats

Results of various treatments groups on body weight of diabetic rats are presented in Figure 2C,D. There was no significant effect on body weight of diabetic rats on days 1 and 5. However, the weight of diabetic control rats (group II) was significantly decreased on day 15 (*p* < 0.001) and 21 (*p* < 0.001) in comparison to the normal saline control rats (group I). Diabetic rats treated with glibenclamide (5 mg/kg, p.o) has significantly (*p* < 0.001) reversed diabetes-induced decrease in body weight on day 15 and 21 compared to diabetic control rats. Moreover, diabetic rats treated with CTN (15, 30 and 50 mg/kg, p.o) reversed the STZ mediated reduction in body weight and significantly increased the body weight on day 15 (*p* < 0.001) followed by day 21 (*p* < 0.001) when compared to diabetic control rats.

#### 2.3.3. Effect of Various Treatments on Liver and Renal Functions in STZ-Induced Diabetic Rats

The activity of hepatic enzymes such as ALP, AST, SGPT, SGOT, and renal functions such as serum creatinine in normal control group, diabetic control group and compound treatment group have been shown in Figure 3. STZ-induced Diabetes in rat considerably increased (*p* < 0.001) the level of liver marker enzymes such as SGPT, SGOT and ALP and renal functions such as serum creatinine and blood urea as compared to the normal control. Administration of standard glibenclamide (5 mg/kg) and CTN (2, 5, 10, 15, 30, and 50 mg/kg body weight) to diabetic rats resulted in significant decrease (*p* < 0.01, *p* < 0.001) in ALP, AST, SGPT, SGOT, serum creatinine, and blood urea levels.

#### 2.3.4. Effect of Various Treatments on Serum Lipid Profile and Hb1c in Diabetic Rats

Figure 4 shows the effects of various treatments on lipid profiles of diabetic rats on day 21. Rats in the diabetic control group (group II) showed significant (*p* < 0.001) increases in plasma lipid parameters TC, TG, LDL, and HbA1c, however a significant (*p* < 0.001) decrease in plasma HDL compared to normal saline control rats (groups I) was observed. Standard glibenclamide (5 mg/kg) has significantly decreased (*p* < 0.001) the plasma TC, TG, and LDL levels in diabetic rats. However, oral administration of CTN (15, 30, and 50 mg/kg, respectively) has significantly decreased (*p* < 0.001) TC, TG, and LDL levels, while significantly increased (*p* < 0.001) HDL cholesterol in diabetic rats was also observed as compared to diabetic control rats.

#### 2.3.5. Effect of Various Treatments on Glutathione Peroxidase

Biochemical analysis showed that antioxidant stress marker enzyme levels such as glutathione peroxidase in liver (Figure 5A) and kidney (Figure 5D) tissue homogenates of STZ-induced diabetic rats were significantly high which were controlled by treatment with berries CTN in STZ-induced diabetic rats suggesting that *E. umbellata* berries possess strong antioxidant proprieties.

#### 2.3.6. Effect of Various Treatments on Lipid Peroxidation and Concentration of Reduced GSH

Figure 5 shows that administration of CTN (2, 5, 10, 15, 30, and 50 mg/kg body weight) has significantly (*p* < 0.05, *p* < 0.00 = 1, *p* < 0.001) reduced MDA level in STZ-induced diabetic rats. The high dose of CTN (50 mg/kg) was more effective as compare to low doses (10 mg/kg and below). CTN treated groups exhibited reduction in lipid peroxidation, observed through significant decrease of MDA level estimated in liver (Figure 5B) and kidney (Figure 5E) tissue homogenates in comparison to the diabetic control group. Furthermore, a significant increase in the concentration of reduced GSH estimated in liver (Figure 5C) and kidney (Figure 5F) tissue homogenates of the CTN treated groups in comparison to diabetic control group was also observed.

#### 2.3.7. Effect of Various Treatments on Pancreas Histopathology in STZ-Induced Diabetic Rats

The effect of CTN administered in different doses on pancreas histopathology in STZ-induced diabetic rats is represented in Figure 6. The histological pattern of the pancreas of the normal control group has shown clear lobular architecture (Figure 6A). Islets of Langerhans were of normal diameter and structure, acinar cells were clear with prominent and centrally placed nuclei and interlobular spaces were also visible. The central cellular integrity and lobular structure of pancreas was retained. The effect of CTN on pancreas histopathology in STZ-induced diabetic rats indicates that a normal histological pattern of the pancreas was shown by the normal control group rats, as presented in Figure 6. Acinar cells were clear with prominent and centrally placed nuclei, and interlobular spaces were also visible. The islet of Langerhans was of normal diameter and structure and the overall cellular integrity and architecture was maintained. The pancreas of Diabetic control group rats (Figure 6B) showed a typical histological pattern with slight grade of congestion and dilated blood vessels with inflammatory cell infiltration. The diameter of islet of Langerhans was decreased with decreased cellular density. Shrined and pyknotic nuclei are also visible in the center of the islet. The boundary of the endocrine and exocrine portion of the pancreas was also indistinct. The presence of inflammatory cells can be observed in connective tissue spaces as well. The histological pattern of the pancreas of group III rats were treated with glibenclamide 5 mg/kg/orally (Figure 6C). The pancreatic tissue of this group of rats did not show any violation from the normal histological pattern during experimental period. No congestion or deterioration was observed in the in acinar cells. There was no reduction in the diameter of Islet of Langerhans and nuclei were also centrally placed. The boundary of endocrine and exocrine portion of pancreas was distinct and overall normal histological pattern was retained. The histological pattern of groups IV and V treated with CTN (2, 5 mg/kg/orally) did not show restoration of pancreatic tissue (Figure 6D,E). The diameter of the islet of Langerhans was declined with reduced cellular mass and the nuclei were not properly aligned. The presence of inflammatory cells and pyknotic nuclei were also observed to some extent, which showed that the CTN with dose 2, 5 mg/kg/orally was not effective to treat STZ-induced diabetes. Group VI-VII (CTN 10, 15 mg/kg/orally) has shown a little bit restoration of histological architecture of pancreas (Figure 6F,G). The size and number of Islet of Langerhans were recovered with low grade rejuvenation but still the distinction of septa between the islets and acinar is not prominent. The presence of inflammatory cells was negligible and the overall cytoplasmic integrity was retained to some extent. The groups VIII–IX (30 and 50 mg/kg/orally) show normal histological pattern of the pancreas with clear lobular architecture (Figure 6H,I). The isslet of Langerhans was of normal diameter and structure, acinar cells were clear with prominent and centrally placed nuclei and interlobular spaces were also visible. The central cellular integrity and lobular structure of the pancreas was also retained. These results showed that the maximum efficacy of CTN to treat diabetes is obtained at a doses of 30 and 50 mg/kg/orally in STZ-induced diabetes.

## 3. Discussion

Diabetes is the world’s fastest growing disease with a high prevalence of morbidity and mortality. This chronic disease is not treatable but can be managed. Consumption of traditional plant medicines and natural product have been in use for centuries to improve diabetic symptoms. In Pakistan, berries fruits are used for the treatment of various disorders including the management of diabetes [13]. So, keeping in view the medicinal and pharmacological importance of *E. umbellata* fruit berries in controlling diabetes the current study was designed to isolate berries catechin and investigate its antidiabetic potential.

Glucose is the end product of carbohydrate metabolism that is most readily absorbed into blood and presented as an increase in post-prandial hyperglycemic condition. Inhibition of carbohydrate digesting enzymes; α-amylase and α-glucosidase can delay the absorption of glucose and consequently results in a decline of post prandial hyperglycemic state. Our results were in line with reported study carried out on *Ononis angustissima* aqueous extract/fraction that exhibited strong inhibitory activity on carbohydrate digestive enzymes; α-amylase and α-glucosidase [23]. In our results berries catechin exhibited strong dose dependent inhibition of digestive enzymes. A similar dose dependent inhibition of α-amylase and α-glucosidase has been observed previously in case of extract/fractions of *Elaeagnus umbellata* [13] and *Salacia oblonga* [24].

The isolated catechin significantly inhibited α-amylase and α-glucosidase enzymes indicating its antihyperglycemic effects, which is in agreement with reported studies on other phenolic compounds [13]. The obtained data confirmed that the use of catechin (2, 5, 10, 15, 30 and 50 mg/kg) and glibenclamide significantly decreased glucose levels in STZ-induced diabetic rats. Studies have shown [25] that the use of glibenclamide increases the sensitivity of β-cells and activates insulin synthesis and secretion from the β-cells of the pancreas that might be associated with its anti-hyperglycemic activity. Furthermore, catechin has protective effects on the major tissues including liver, kidney, and pancreas thus have a minimizing effect of diabetes associated complications [13].

It was also observed that catechin in the dose of 30 and 50 mg/kg body weight can significantly improve the adverse metabolic effects in the serum of animals treated with streptozotocin. STZ-induced diabetic rats treated with catechin have significantly reduced serum glucose level, improved lipid profiles (TC, TGs, LDL, HDL, and cholesterol), liver profiles (SGPT, SGOT, ALP, AST, and serum creatinine), and body weight. The results of the current study confirm the previous outcomes reported by other researchers that catechin has significantly improved STZ damages in rats [21,22,26]. The current study also indicates the improvement in oxidative stress in the STZ diabetic rats that might be due to strong antioxidant potential of catechin against DPPH and ABTS free radicals. Our results are in line with the reported study that hyperglycemia in STZ treated group is due to the cytotoxic effect of STZ on insulin secreting β-cells of pancreas that are also accompanied with an increase in lipid peroxidation. Thus, destruction of pancreatic β-cells is related to the locally and systemically induced oxidative stress [13,27,28,29].

A significant increase in lipid parameters levels such as TC, TGs, LDL, and cholesterol was observed while at the same time a significant decrease of HDL cholesterol in STZ-induced diabetic rats was also evident in comparison to normal control group. These results were in line with other previous studies where it has been observed that increase in circulating free fatty acids enhances hepatic TGs production. High TGs content associated with formation of an atherogenic dyslipidaemia consisting of an elevation in high TG content, low HDL cholesterol and increased LDL or apo lipoprotein-B [19,30]. Epidemiological studies indicated that regular intake of more than two cups of green tea and its catechin have been shown to ameliorate oxidative stress, hyperlipidemia and lower the concentration of TC and LDL, which has also been demonstrated in the performed meta-analysis of randomized clinical trials [31].

It was also observed that STZ-induced diabetic rats showed a considerable increase in the levels of hepatic enzymes such as SGPT, SGOT, ALP, AST, and serum creatinine. The improvement of STZ effects in rats after treatment with catechin might indicate a protective effect of it against STZ function due to decrease in oxygen free radicals production and also increase in antioxidant content. In the present investigation, SGPT, SGOT, ALP, AST, and serum creatinine activities were significantly reduced in catechin treated diabetic rats in comparison to the untreated diabetic rats which were in line with previous findings [13,31]. It has been demonstrated in a related study the hypoglycemic and antioxidant activity of catechin in the STZ-induced diabetic rat are also associated with reduction in lipid peroxidation [26].

The observed high level Hb1Ac; the key biomarker of diabetes in STZ-induced diabetic rat model also in line with reported studies [8,32]. The HbA1c value may greatly be affected by variations in the amount of glucose that enters the erythrocyte membrane, either to increase the rate of glycosylation or change the lifespan of erythrocytes resulting in an increase or decrease in HbA1c value, respectively [8].

Among phenolic compounds; flavonoids have been reported as scavenger of free radicals that shows strong antioxidant activity by donating hydrogen group. Catechin also exhibits its antioxidant and free radicals scavenging activity through hydrogen donating ability in STZ-diabetic rats. Glutathione peroxidase is the key antioxidant enzyme which plays a major role in oxidative damage against oxidative stress induced by free radicals. The current study showed that streptozotocin administration in rat’s results in decreased activities of glutathione peroxidase which is a validation of the other investigations. Catechin treatments of diabetic rats cause reduction in free radical generation and lipid peroxidation thus elevating antioxidant defence. Antioxidant activity of catechin has been validated in several other studies as well [19,26].

Streptozotocin causes lipid peroxidation and increases the concentration of MDA in hepatocytes. Increases in MDA level is an important indicator of tissue damages. Administration of berries catechin (2, 5, 10, 15, 30, and 50mg/kg body weight) markedly reduced the MDA content near to normal that has been validated by other studies on plant extracts [6]. Antioxidant enzymes such as glutathione peroxidase are involved in the reduction of free radical and oxidative stress. STZ intoxicated rats exhibited a decrease in the total content of reduced GSH level, GPx activities, and elevated MDA level. Increased MDA level have been shown to be an important marker for in vivo lipid peroxidation. Oxidative stress results from a marked imbalance between free radical production and elimination by antioxidant system. Furthermore, berries catechin (2, 5, 10, 15, 30, and 50 mg/kg body weight) significantly (*p* < 0.05, *p* < 0.01, *p* < 0.001) increased GPx activities and the content of reduced GSH in both liver and kidney homogenates, and decreased MDA level. Such regulation of oxidative stress markers by the catechin may be well correlated with previous reports, where catechin have antioxidant and free radical scavenging ability [31].

A significant decrease in body weight was also observed after 21 days in STZ- induced diabetic rats. It is majorly due to decrease of carbohydrate reserve as an energy source and fats catabolism during diabetic conditions that usually cause excessive break down of tissue protein that leads to loss in body weight. However, oral administration of *E. umbellata* berries catechin at a dose range of 30 and 50 mg/kg body weight for 21 days to diabetic rats brings a pronounced increase in body weight. Similar effects have also been reported in other studies where restoration in body weight with extract of *Ginkgo biloba* [33], *Afzelia africana* [34], and *Elaeagnus umbellata* [13] have been recorded in STZ-induced diabetic rats. These results designate that *E. umbellata* berries possessed the ability of body weight restoration and management of glucose levels.

Histopathological analysis clearly showed improvement in pancreas section of the treated groups (catechin treated groups) when compared to STZ-induced diabetic group. The pancreatic tissue of diabetic rat demonstrated degeneration and vacuolization in the Langerhans’s islet cells. The diameter of the islet of Langerhans was decreased with decreased cellular density. Shrined and pyknotic nuclei were also visible in the center of the islet. The boundary of endocrine and exocrine portion of the pancreas was also indistinct, besides decrease in islets size and β-cell number was also observed. This data was in agreement with the study of Ramadan et al. [35]. Current results indicated that histological pattern of pancreas of groups treated with glibenclamide (5 g/kg/overly) and catechin (15, 30, and 50 mg/kg; p.o) did not show any violation from the normal histological pattern during experimental period. Similar to previous investigations, our study demonstrated that glibenclamide and catechin treatment are useful to increase pancreatic regeneration, number and diameter of islet of Langerhans and an increase in β-cell function after treatment in STZ-induced diabetic rates [36].

In conclusion, the current study designated that the anti hyperglycemic and hypolipidemic ability of catechin exhibits a protective effect on various parameters in the STZ induced diabetic rats and helped in relieving the related complications. Based on the results of the histopathological studies of the pancreatic tissue, the anti-diabetic activity of those samples was believed to occur through the mechanism of increasing secretion of healthy pancreatic β-cells. However, more studies are needed for evaluating the exact protective mechanism of catechin for diabetes management and its complications in animal models.

## 4. Materials and Methods

### 4.1. Chemicals and Reagents

Antioxidant chemicals such as DPPH, ABTS, and ascorbic acid were purchased from Sigma-Aldrich (St. Louis, MO, USA); while antidiabetic chemicals and reagents such as Type-I α-Glucosidase (Baker Yeast), Type-VI α-amylase (porcine pancreas), PNPG (*p*-nitrophenyl-α-d-glucopyranose), streptozotocin were obtained from Sigma Aldrich (Darmstadt, Germany); glucose estimation kits from SD Chek-Gold Germany, and glibenclamide was provided by Sanofi-Aventis-Pharma Pakistan. Chemicals such as Tween-80 (Scharlau-chem., Barcelona, Spain), normal saline solution (Utsoka Pharma, Las Bela Baluchistan, Pakistan), lipid profile tests kits (Human, Hamburg, Germany), and renal profile tests kits, antioxidant enzymes kits (Biomed: Germany; diagnostic) were used in in vivo study. Various solvents such as methanol, *n*-hexane, ethyl acetate, and chloroform (Merck, Darmstadt, Germany) were also used in this study.

### 4.2. Plant Material Collection

*E. umbellata* berries fruits were collected from the mountainous regions of District Kalam, Malakand Division, Khyber Pakhtunkhwa, Pakistan, in the month of August and September 2019. The plant specimen was identified at Post Graduate College Swat, Khyber Pakhtunkhwa, Pakistan. Voucher specimen (BGH-UOM-154) were deposited in the Botanical Garden Herbarium, University of Malakand, Pakistan [13].

### 4.3. Extraction, Fractionation and Isolation of Pure Compound

According to the method presented in a previously published article [13], the plant extract preparation and fractionation were carried out. About 10 kg of dried berries fruits were crushed through mechanical grinder to obtain fine powders. The extraction was carried out using fruit powder mixed in 80% methanol for two weeks with periodical shaking. The resulting mixture was filtered using muslin cloth followed by Whattman filter paper and the filtrate was dried into a semisolid mass at 45 °C under reduced pressure in the rotary evaporator. Finally the resultant 750 g semisolid mass of extract was solidified in an open air. The resultant methanol extract in the specified amount was dissolved in a mixture (0.5 L) of methanol/water (7:3) and successively extracted by solvent–solvent extraction method using different solvents (*n*-hexane, chloroform, ethyl acetate *n*- butanol, and aqueous starting from a low polarity to high. On the basis of in vitro analysis, the result of the chloroform fraction was found as the most active and was further subjected to isolation of bioactive compounds. The chloroform fraction was subjected to silica gel chromatography column with internal diameter of 10 cm packed height 50 cm in solvent system of *n*-hexane and ethyl acetate (3:7) yielded compound 2.5 g (catechin) in pure form. The compound was characterized through different spectroscopic techniques such as FTIR, H-NMR, ^13^C-NMR, and Mass spectrometry. The purified compound catechin was screened for in vitro and in vivo antioxidant and antidiabetic activity.

### 4.4. Structural Confirmation and Characterization of Isolated Compounds

The structure of the isolated catechin (2-(3,4-dihydroxyphenyl)-3,4-dihydro-2H-chromene-3,5,7-triol) is presented in Appendix A. The molecular formula of catechin was determined as C_15_H_14_O_6_ by ESIMS and Molar mass (*m*/*z*) is 290.1 g/mol. ^1^H-NMR (METHANOL-d_4_,300MHz): δ = 6.73–6.62, signifies the aromatic protons at 15th, 20th and 21th carbon of the given structure, δ = 5.83 (s, 1H) this singlet could be the hydrogen atom at carbon number 11, δ = 5.76 (s, 1H) this singlet could be attributed to proton attached with carbon number 8 of the given structure, 4.76 (s, 4H) this broad singlet represents the four protons each one attached as a –OH group to the carbon number 6, 9, 16 and 18. Δ = 4.46 (d, *J* = 7.5 Hz, 1H), 3.87 (d, *J* = 5.1 Hz, 1H) this duplet can be precisely represent by a proton attached to the carbon number 2 of the given structure, δ = 2.71 ppm (s, 1H) this singlet has been contributed by the protons attached to the carbon number 4 of the given structure (Appendix A). ^13^C-NMR (METHANOL-d_4_, 75MHz): δ = 156.4, 156.2, 155.5, 144.8, 130.8, 118.6, 113.9, 99.4, 94.9, 94.1, 81.5, 67.4, 27.1 ppm (Appendix A). The EIMS and FTIR spectrum is represented in Appendix A.

### 4.5. DPPH (2,2-diphenyl-1-picrylhydrazyl) Free Radical Scavenging Assay

DPPH (2,2-diphenyl-1-picrylhydrazyl) free radicals scavenging potential of catechin was determined using Brand-Williams assay [37] with some modification. DPPH solution was prepared in methanol (100 mL). 1 mg/mL catechin was also prepared in methanol with serial dilutions 31.05, 62.5, 250, 500, and 1000 µg/mL. Catechin (0.1 mL) was mixed with DPPH (3.0 mL) and incubated for 30 min at 23 °C. Finally absorbance was measured at 517 nm using UV spectrophotometer and ascorbic acid was used as a positive control. Results were taken in three replicates and represented as mean ± SEM. Percent DPPH scavenging potential was calculated using the following Equation (1):
(1)
%Free radicals scavenging potential =  Blank sample absorbance−sample absorbanceBlank sample absorbance×100. 


### 4.6. ABTS (2,2′-azinobis-3-ethylbenzothiazoline-6-sulfonic acid) Free Radical Scavenging Assay

ABTS (2,2′-azinobis-3-ethylbenzothiazoline-6-sulfonic acid) free radicals were produced according to the reported assay [38]. Antiradical potential of catechin was measured by taking 300 µL catechin solution and 3.0 mL ABTS solution, mixed thoroughly and incubated for 6 min. Finally the absorbance was measured by UV spectrophotometer. Ascorbic acid as positive control as mentioned above have been used. Percent ABTS free radicals scavenging potential was calculated using Equation (1).

### 4.7. In Vitro α-Amylase Inhibitory Assay

The α-amylase inhibitory potential was estimated using 3,5-dinitrosalicylic acid (DNSA) assay [39]. Isolated compound catechin was dissolved in mixture of DMSO (10%), Na_2_HPO_4_/NaH_2_PO_4_ buffer (0.02M) and NaCl (0.006M) at pH 6.9 and then various dilutions were prepared in the range 31.05, 62.5, 125, 250, 500 and 1000 μg/mL. Enzyme α-amylase (200 μL; 2 units/mL) was mixed with catechin (200 μL) and incubated for 10 min at 30 °C. Then 200 μL starch (1%) solution was added to each serial dilution and incubated for 3 min. To stop the reaction 200 μL sodium potassium tartrate tetrahydrate (DNSA) reagent, 2M NaOH, and 20 mL of 3,5 dinitrosalicylic acid (96 mM) were added to the reaction mixture. The reaction mixtures were boiled at 85–90 °C for 10 min in a water bath. After cooling the mixture was diluted with distilled water (5 mL) and then the absorbance was recorded at 540 nm through UV Visible spectrophotometer. A blank solution was prepared containing only isolated compound catechin but no enzyme. Standard acarbose (100 μg/mL–2 μg/mL) was prepared as positive control. The α-amylase enzyme inhibitory activity was calculated using the Equation (2):
(2)
%α−amylase Inhibition = control absorbance−sample absorbancecontrol absorbance×100


### 4.8. In Vitro α-Glucosidase Inhibitory Assay

The α-glucosidase inhibition of isolated compound catechin was assessed according to the reported method [40] with minor modifications. The reaction mixture was prepared by adding 100 μL α-glucosidase (0.5 unit/mL) enzyme, 600 μL phosphate buffer (0.1 M; pH 6.9), and 50 μL compound dilutions (31.05, 62.5, 125, 250, 500 and 1000 µg/mL). The reaction mixture was incubated at 37 °C for 15 min. The enzymatic reactions were started by adding 100 μL substrate (*p*-nitro-phenyl-α-d-glucopyranoside) solution (5 mM) prepared in phosphate buffer (0.1 M; pH 6.9), then incubated for 15 min at 37 °C. The reaction was stopped by adding 400 μL sodium carbonate (0.2 M) solution and finally absorbance was recorded at 405 nm. The reaction mixture without compound was used as a control and the blank solution was prepared without α-glucosidase enzyme. Percent α-glucosidase inhibitory potential was calculated by the following Equation (3):
(3)
%α−Glucosidase Inhibition =control absorbance−sample absorbancecontrol absorbance×100


### 4.9. Animals

The in vivo experiments were conducted on Sprague-Dawley adults albino rats having weight from 170–200 g that were purchased from Rifah Institute of Pharmaceutical Sciences Islamabad, Pakistan. The animals were maintained at 22–25 °C with food and fresh water ad libitum. The animals were acclimated to animal house conditions at room temperature around 22–25 °C. The animals were housed three rats per cage with light and dark cycle of about 12 h. 

### 4.10. Acute Toxicity Study

All the animals were divided into nine groups that were comprising of 8 animals each. The control group animals received tween-80 suspension, orally. All animals were then treated orally with different doses (50, 100, 150, 200, 250, 300, and 500 mg/kg body weight) of CTN for acute toxicity study. Immediately after dosing, the animals were observed continuously for 4 h for symptoms of toxicity (respiratory aches, motor activity, convulsions, muscle spasm, tremors, sedation, diarrhoea, lacrimation, hypnosis, salivation, and loss of righting reflex) in animals. All animals were kept under observation and seemed healthy at 24 h to 1 week with no obvious alterations in appearance or behavior. No mortality was observed up to one week. The CTN remained safe and nontoxic up to 500 mg/kg body weight dose range. Therefore, according to Organization for Economic Cooperation and Development (OECD) guidelines, CTN at dose 50 mg/kg body weight as 1/10th of the highest dose (500 mg/kg) for multiple dose investigation was selected to evaluate in vivo antidiabetic activity of CTN [41].

### 4.11. Animal Experimental Design for Induction of Diabetes

Diabetes was induced according to the method previously described by Matsuzawa-Nagappa et al. [42]. Animals were divided into two major groups. One group was given normal pellet diet and the other animal group was fed with high fat diet (HFD: 40% raw beef fat + 30% casein + 10% glucose + 7% wheat flour + 6% barn + 4% vitamin mixture and 3% salt mixture) for two weeks before commencing the experiment. After two weeks, induction of hyperglycemia was carried out in HFD Sprague-Dawley rats via a single intraperitoneal (i.p) injection of STZ (50 mg/kg) prepared in 0.9% normal saline solution after an overnight fast. Blood samples were collected from tail vein and through SD glucometer blood glucose level was measured after 72 h (of STZ injection). Rats with blood glucose levels higher than 300 mg/dL were considered as being diabetic. In the normal control groups only normal saline (8 mL/kg, p.o) was given as a vehicle. Isolated compound catechin was administered to the treatment groups from 3 days after STZ administration for 3 weeks. Blood glucose level and body weights were documented at weekly intervals.

### 4.12. Detailed Treatment Protocol

Rats were divided into 9 groups (*n* = 8) after an overnight fast for about 12 h. The first group labeled as normal control that takes normal saline orally while the rest of eight groups were considered HFD groups. In this study the STZ-induced diabetic model was used to evaluate the anti-hyperglycemic and anti-hyperlipidemic activity of isolated compound catechin (CTN) in chronic multiple dose experiment. The schematic treatment was as follows:

Group I: Normal control animals received normal saline (8 mL/kg, p.o)

Group II: Diabetic control received STZ (50 mg/kg, i.p.)+ Normal saline, p.o

Group III: Standard control group animals received STZ (50 mg/kg, i.p.) + glibenclamide (5 mg/kg, p.o)

Group IV: Treated group animals received STZ (50 mg/kg, i.p.) + CTN (2 mg/kg, p.o)

Group V: Treated group animals received STZ (50 mg/kg, i.p.) + CTN (5 mg/kg, p.o)

Group VI: Treated group animals received STZ (50 mg/kg, i.p.) + CTN (10 mg/kg, p.o)

Group VII: Treated group animals received STZ (50 mg/kg, i.p.) + CTN (15 mg/kg, p.o)

Group VIII: Treated group animals received STZ (50 mg/kg, i.p.) + CTN (30 mg/kg, p.o)

Group IX: Treated group animals received STZ (50 mg/kg, i.p.) + CTN (50 mg/kg, p.o)

All the animals were given their respective treatments for 21 days starts daily at 09:00 am. The fasting blood glucose levels were determined at 0, 4th, 7th, 10th, 15th and 21st days in fasting condition with the glucometer from the tail vein. The same protocol was repeated once again. Changes in the body weight of untreated control and experimental animals were documented in fasting condition at day 1 and after 21 days of experiment and are presented as % increase in body weight [13].

### 4.13. Collection of Blood and Estimation of Biochemical Parameters

At the completion of in vivo antidiabetic activity on the 21st day, all animals were anesthetized by intraperitoneal (IP) injection of 35 mg/kg pentobarbital sodium and euthanized by means of cervical decapitation according to the previously reported procedure illustrated in schedule-1 of UK, animal scientific procedure act; 1986. Blood collection was carried out through cardiac puncture, transferred to EDTA and non-EDTA containing tubes. Blood and sera were separated by centrifugation at 3500 rpm for 10 min for analysis of glucose and biochemical parameters such as serum alkaline phosphatase (ALP), aspartate aminotransferase (AST), serum glutamate pyruvate transaminase (SGPT), and serum glutamate oxaloacetate transaminase (SGOT) [41]. The concentration of lipid profile parameters; total cholesterol (TC), triglycerides (TG), low-density lipoproteins (LDL), high-density lipoprotein (HDL), and serum creatinine in serum were estimated using a biochemistry analyzer (PS-520; Shenzhen Procan Electronics, China) as per companies’ instructions using diagnostic kits (Reactivos, GPL Barcelona, Spain) [41]. The level of glycosylated hemoglobin (HbA1c) was performed by an enzyme-linked immunosorbent assay (ELISA) kit according to the reported protocol [7].

### 4.14. Measurement of Antioxidant Enzymes and Biochemical Parameters in Hepatic and Kidney Tissues

The liver and kidney tissue were removed, washed with cold saline solution and blotted dried. A weighed portion of liver and kidney portion were homogenized and 10 percent homogenate (*w*/*v*) was prepared in EDTA (1mM) and phosphate buffer (100mM) solution. Homogenates were placed at 4 °C for 20 min, and finally centrifuged at 3500 rpm for 15 min at 4 °C and supernatants were divided into aliquots and stored at −20 °C until assessed for different enzymes. According to the reported method [43] activities of glutathione peroxidase (GPx) and total content of reduced glutathione (GSH) level in the supernatant obtained from liver and kidney tissues were analyzed. Thiobarbituric acid reactive substance (TBARS) assay was used for estimation of and malondialdehyde (MDA) level (lipid peroxidation) [6].

### 4.15. Histopathology

For histopathological examination the pancreas portion were immediately removed, washed thoroughly with saline solution to remove the blood. Finally the pancreas tissue were fixed in 10% formalin, dehydrated with ethanol-xylene mixtures and fixed with paraffin. Tissue blocks were sectioned with about 4.5–6.5 μm thickness through Microtome (ACCU-Cut^®^ SRM^™^: 200 Sakura). Slides were stained with Hematoxylin and Eosin (H and E) dye by means of automatic slide stainer (Sakura Tissue-Tek^®^ DRS^™^ 2000; Salt Lake Utah, USA). The stained slides were cleaned properly and observed under microscope to see alteration in pancreatic tissues (architecture of STZ-induced diabetic group) and treated (glibenclamide/extract) groups. The samples were processed according to reported protocols [43].

### 4.16. Statistical Analysis

All in vitro and in vivo experiments were performed in three replicates. All results were determined as mean ± SEM. The Student’s *t*-test and one way ANOVA followed by Dunnett’s post hoc multiple comparison test was used. *p* ≤ 0.05 were considered as significant. Linear regression was used to calculate IC_50_ for % DPPH, ABTS, α-amylase, and α-glucosidase inhibition against the different concentration of test samples by means of Excel program 2007.

## 5. Conclusions

The observed post-prandial hypoglycemic potential of *E. umbellata* catechin can be attributed to its inhibitor effects on carbohydrate digestive enzymes (α-amylase and α-glucosidase). CTN played a significant role in the amelioration of oxidative stress and diabetic complications in STZ-induced injuries in liver and kidneys in rats. CTN significantly decreased hepatic, renal, and lipid parameters, and HbA1c levels in diabetic rats while significantly increased the HDL level, antioxidant enzyme GPx and reduced GSH content. It can be concluded from the results of this study that *E. umbellata* berries catechin could effectively be used as an antioxidant and anti-hyperglycemic factor that has potential to reduce the MDA level in STZ-induced diabetic rats, and thus might be a useful candidate to be used in the management of type-2 diabetes.

## Figures and Tables

**Figure 1 molecules-26-00137-f001:** Percent in vitro free radicals (DPPH and ABTS) scavenging and antidiabetic (α-amylase and α-glucosidase) activity of *E. umbellata* berries catechin. (**A**) % DPPH scavenging activity, (**B**) % ABTS scavenging activity, (**C**) percent α-amylase inhibition potential, (**D**) α-glucosidase inhibition potential of *E. umbellata* berries catechin at various concentrations.

**Figure 2 molecules-26-00137-f002:** Effect of various treatments on blood glucose level and body weight of STZ-induced diabetic rats (**A**,**B**). Blood glucose level in different treatment groups (**C**,**D**). Effect of various treatment groups on body weight in STZ-induced diabetic rats. (**A**) is showing the blood glucose levels at the end of treatment period while (**B**) shows blood glucose levels at different week. (**C**) is showing the body weight (mg/kg) at the end of treatment period while (**D**) shows body weight (mg/kg) at different week. [The values are expressed as mean ± SEM. ns = non-significant. Each value corresponds to a mean of eight animals. **^###^**
*p* ˂ 0.001; comparison of ^a^(normal control) vs. ^b^(diabetic control) using Student’s *t*-test, * *p* ˂ 0.05, ** *p* ˂ 0.01, *** *p* < 0.001; comparison of ^b^(diabetic control) vs.^c^ (glibenclamide and CTN treated groups) using one way ANOVA followed by Dunnett’s post hoc multiple comparison test].

**Figure 3 molecules-26-00137-f003:** Effects of catechin on the level of liver serum biochemical parameters in STZ-induced diabetic rats: (**A**) Serum level of ALP, (**B**) serum level of AST, (**C**) serum level of SGPT, (**D**) serum level of SGOT, (**E**) serum level of creatinine in STZ-induced diabetic rats. The values are expressed as mean ± SEM. Each value corresponds to a mean of eight animals ^###^
*p* < 0.001; comparison of ^a^(normal control) vs. ^b^(diabetic control) using Student’s *t*-test, ** *p* ˂ 0.01, *** *p* < 0.001; comparison of ^b^(diabetic control) vs. ^c^(Glibenclamide and CTN treated groups) using one way ANOVA followed by Dunnett’s post hoc multiple comparison test.

**Figure 4 molecules-26-00137-f004:** Effects of catechin on the level of serum lipid parameters and Hb1c level in STZ-induced diabetic rats (**A**) Serum level of TC, (**B**) serum level of TG, (**C**) serum level of LDL, (**D**) serum level of HDL, (**E**) serum level of Hb1c in STZ-induced diabetic rats. The values are expressed as mean ± SEM. Each value corresponds to a mean of eight animals ### *p* ˂ 0.001; comparison of ^a^(normal control) vs. ^b^(diabetic control) using Student’s *t*-test, * *p* ˂ 0.05, ** *p* ˂ 0.01, *** *p* < 0.001; comparison of ^b^(diabetic control) vs. ^c^(glibenclamide and CTN treated groups) using one way ANOVA followed by Dunnett’s post hoc multiple comparison test.

**Figure 5 molecules-26-00137-f005:** Effects of catechin (2, 5, 10, 15, 30, and 50 mg/kg) and glibenclamide (5 mg/kg) on liver and kidney oxidative stress markers in diabetic rat in STZ-induced diabetic rats. Oxidative stress markers in liver: (**A**) GPx, (**B**) malondialdehyde (MDA) (**C**) glutathione (GSH). In kidney: (**D**) GPx, (**E**) MDA, (**F**) GSH. The values are expressed as mean ± SEM. Each value corresponds to a mean of eight animals ^###^
*p* ˂ 0.001; comparison of ^a^(normal control) vs. ^b^(diabetic control) using student *t*-test, * *p* ˂ 0.05, ** *p* ˂ 0.01, *** *p* < 0.001; comparison of ^b^(diabetic control) vs. ^c^(Glibenclamide and CTN treated groups) using one way ANOVA followed by Dunnett’s post hoc multiple comparison test.

**Figure 6 molecules-26-00137-f006:** Effects of catechin (2, 5, 10, 15, 30 and 50 mg/kg) and glibenclamide (5 mg/kg) on pancreas histopathology studies in STZ-induced diabetic rats (H&E staining; 40×, scale bar = 100 μm): (**A**) control group; (**B**) diabetic control group; (**C**) glibenclamide (5 mg/kg/overly) treated group; (**D**,**E**) CTN (2, 5 mg/kg/orally) treated group; (**F**,**G**) CTN (10, 15 mg/kg/orally) treated groups (**H**,**I**) CTN (30 and 50 mg/kg/orally) treated group.

## Data Availability

The data presented in this manuscript belong to the PhD research work of Mrs. Nausheen Nazir and has not been deposited in any repository yet. However, the data are available to the researchers upon request.

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
