# Peer review of "Curative Effect of Catechin Isolated from Elaeagnus Umbellata Thunb. Berries for Diabetes and Related Complications in Streptozotocin-Induced Diabetic Rats Model"

_molecules, 2020, doi:10.3390/molecules26010137_

Round 1
Reviewer 1 Report
Reviewer Comments
The current study was executed to isolation berries catechin and investigation antidiabetic potential of natural flavonoid and Glibenclamide. The manuscript is written compact and easy to follow. The introduction provides sufficient, relevant background informations. Experimental methods are suitable used and generally appropriate. Obtained findings are useful.
Several specific comments are as follow:
Line 128: What purify of isolated catechin was achieved?
Line 150: Please correct typing error.
Line 190: What is the laboratory rats diet composition? Were rats maintained on ST-1?
Line 202-209: Only type 2 diabetes mellitus is mentioned in this publication as well as Glibenclamide is drug used in the treatment of type 2 diabetes. In contrast, streptozotocine administered at dose levels 50 mg/kg is commonly used as a rodent model of type 1 diabetes mellitus.
Line 296: Please correct typing error. I assume that ascorbic acid is used as positive control.
Line 279: Catechin exhibited significant antiradical and inhibitory potential at concentration of 1000 µg/ml in the case of in vitro experiment. The lower doses don´t show significant antiradical and inhibitory potential?
Line 307-309: What do they mean the values shown in Figure 2A (marked Catechin 2-10)? Does it means a group of animals that received STZ+CTN? If so, significant effect is presented at dose of 2-10 mg/kg, compared to diabetic control rats. If so, please fill in Figure 2A according to study design.
Line 337-338, 379-380: Administration of CTN (2-10 mg/kg) to diabetic rats resulted in significant decline or growth as well.
Line 488: Specify the dose range at which catechin lowers blood glucose.
Line 540-541: Specify the dose range at which catechin reduced MDA content.
Line 633: Please write the name of journal in italics.
Line 665: Please correct the autor´s name from a formal point of view.
Line 657: Please correct the punctuation at the end of the line.
Line 663-664: Please correct the citation.
Line 687: Please correct the citation.
Line 689: Please correct and complete citation.
Line 702: Please complete citation.
Line 705: Please correct and complete citation.
Author Response
Reviewer 1
The current study was executed to isolation berries catechin and investigation antidiabetic potential of natural flavonoid and Glibenclamide. The manuscript is written compact and easy to follow. The introduction provides sufficient, relevant background information’s. Experimental methods are suitable used and generally appropriate. Obtained findings are useful.
Dear Editor/ reviewer,
Thank you very much for kind review and comments concerning our manuscript. Thank you so much worthy editors/reviewers for appreciation the manuscript data, experimental method used/plan of study, and about the findings of study. We appreciate the hard work of reviewers as they fairly pointed out errors and mistakes in our manuscript. We have tried to revise the manuscript in line with comments of the reviewers. Corrections made have been highlighted as Blue.
Please find below the point by point responses to the reviewer’s comments and suggestions.
Several specific comments are as follow:
Line 128: What purify of isolated catechin was achieved?
- Answer: Worthy reviewer the purity of purified catechin was about 97.4%.
Line 150: Please correct typing error.
- Answer: Worthy reviewer the statement has been corrected accordingly.
Line 190: What is the laboratory rats diet composition? Were rats maintained on ST-1?
- Answer: Worthy reviewer animal in treated groups were fed with high fat diet (HFD) (40% raw beef fat + 30% casein + 10% glucose + 7% wheat flour + 6% barn + 4% vitamin mixture and 3% salt mixture) for two weeks before commencing the experiment while normal control group throughout and treated groups after 2 weeks were maintained on normal pellet diet containing Dried skim milk, fish meal, soybean meal, corn gluten meal, soybean oil and salt as major constituents while minor constituents are different minerals and vitamins
- Worthy reviewer all these information has also been included in the revised manuscript. Normal pellet diet was given to control group
Line 202-209: Only type 2 diabetes mellitus is mentioned in this publication as well as Glibenclamide is drug used in the treatment of type 2 diabetes. In contrast, streptozotocin administered at dose levels 50 mg/kg is commonly used as a rodent model of type 1 diabetes mellitus.
- Answer: Worthy reviewer this current investigational study was designed for the treatment of type 2 diabetes and Glibenclamide drug is used in the treatment of type 2 diabetes. As we know that streptozotocin administered at dose levels 50 mg/kg is commonly used as rodent model of type 1 diabetes but it is also reported to use as rodent model of type 2 diabetes (T2D) as well [1].
- Worthy reviewer in this study we use high-fat diet (HFD) and Streptozotocin (STZ) to induce T2D in animal model. The use of high-fat diet (HFD) and Streptozotocin (STZ) to induce T2D in rats has already been reported in the literature [2-4]. In this model, an administration of HFD causes obesity in rats which leads to insulin resistance. Insulin resistance is a key pathophysiological feature of T2D.
- Furthermore, low dose of STZ which is known as diabetogenic a β- cell toxin, causes destruction and severe decline of β- cells [5, 6]. As a result, lacking of insulin also causes hyperglycemia [7]. Thus the hyperglycemia coupled with other metabolic irregularities including insulin resistance and hyperlipidemia closely depict the metabolic appearances of T2D [8, 9].
Line 296: Please correct typing error. I assume that ascorbic acid is used as positive control.
- Answer: Worthy reviewer ascorbic acid is used as a positive control in antioxidant activity. Worthy reviewer the statement given in the comment belong to the section in vitro antidiabetic activity in which acarbose is used as a positive control.
Line 279: Catechin exhibited significant antiradical and inhibitory potential at concentration of 1000 µg/ml in the case of in vitro experiment. The lower doses don´t show significant antiradical and inhibitory potential?
- Answer: Respected reviewer catechin exhibited significant inhibition potential against DPPH and ABTS at concentration range of 1000, 500, 250, 125, 62.5, and 31.25µg/mL. Worthy reviewer to reduce confusion the significant levels were removed from the Figure 1 (A&B) and added Table S1 in Supplementary file. The data related the Table S1 has also been added in the revised manuscript.
Line 307-309: What do they mean the values shown in Figure 2A (marked Catechin 2-10)? Does it means a group of animals that received STZ+CTN? If so, significant effect is presented at dose of 2-10 mg/kg, compared to diabetic control rats. If so, please fill in Figure 2A according to study design.
- Answer: Worthy reviewer the in Figure 2A the animal groups marked as catechin 2-10 means the groups of animals that received STZ+CTN. Worthy reviewer the Figure 2A has been revised and correct the animal groups according to the study design. Also the text explain these facts has been revised accordingly
Line 337-338, 379-380: Administration of CTN (2-10 mg/kg) to diabetic rats resulted in significant decline or growth as well.
- Answer: Worthy reviewer all the treatments groups of CTN exhibited significant decline in the studied parameters and has been added in the revised manuscript accordingly. There were mistakes in these sentences which were corrected accordingly.
Line 488: Specify the dose range at which catechin lowers blood glucose.
- Answer: Respected reviewer the all tested doses (2, 5, 10, 15, 30 and 50 mg/kg) lowered the blood glucose level (fasting blood glucose level of diabetic rats were significantly (P<0.01, P<0.001) decreased compared to diabetic control rats). However, from 15 to 50 mg/kg body weight the effect was more pronounced. Blood glucose level reduction was observable from the 5th day and onward and the patterns were apparent on days 10 (P<0.05), 15, and 21 (P<0.001).
Line 540-541: Specify the dose range at which catechin reduced MDA content.
- Answer: worthy reviewer all the tested doses reduced the MDA level as is clear from figure 5 however, the effect was more pronounced for high doses.
Line 633: Please write the name of journal in italics.
- Answer: Corrected accordingly.
Line 665: Please correct the autor´s name from a formal point of view.
- Answer: Corrected accordingly.
Line 657: Please correct the punctuation at the end of the line.
- Answer: Corrected accordingly.
Line 663-664: Please correct the citation.
- Answer: Corrected accordingly
Line 687: Please correct the citation.
- Answer: Corrected accordingly
Line 689: Please correct and complete citation.
- Answer: Corrected accordingly
Line 702: Please complete citation.
- Answer: Corrected accordingly
Line 705: Please correct and complete citation.
- Answer: Corrected accordingly
- Worthy reviewer in the recommended citation given below the year and the volume is both 2013.
- Cheng, D.; Liang, B.; Li, Y. Antihyperglycemic effect of Ginkgo biloba extract in streptozotocin-induced diabetes in rats. Biomed. Res. Int. 2013, 2013, 162724.
References:
- Nazir, N.; Zahoor, M.; Nisar, M.; Khan, I.; Karim, N.; Abdel-Halim, H.; Ali, A. Phytochemical analysis and antidiabetic potential of Elaeagnus umbellata (Thunb.) in streptozotocin-induced diabetic rats: pharmacological and computational approach. BMC Complement. Altern. Med. 2018, 18(1), 332.
- Matsuzawa-Nagata, N.; Takamura, T.; Ando, H.; Nakamura, S.; Kurita, S.; Misu, H.; Ota, T.; Yokoyama, M.; Honda, M.; Miyamoto, K.; Kaneko, S. Increased oxidative stress precedes the onset of high-fat diet-induced insulin resistance and obesity. Metabolism. 2008, 57(8), 1071-7.
- Skovso, S. Modeling type 2 diabetes in rats using high fat diet and streptozotocin. J Diabetes Investig. 2014, 5(4), 349-358.
- Parveen, K.; Khan, R.; Siddiqui, AW. Antidiabetic effects afforded by Terminalia arjuna in highfat-fed and streptozotocin-induced type 2 diabetic rats. Int J Diab Metab. 2011, 19, 23–33.
- Lenzen, S, . The mechanisms of alloxan-and streptozotocin-induced diabetes. Diabetol. 2008, 51, 216–226.
- Reed, MJ.; Meszaros, K.; Entes, LJ.; Claypool, MD.; Pinkett, JG.; Gadbois, TM.; Reaven, GM. A new rat model of type 2 diabetes: The fat-fed, streptozotocin-treated rat. Metab. Clin. Exp. 2000, 49(11), 1390-1394.
- Grover, JK.; Yadav, S.; Vats, V. Medicinal plants of India with antidiabetic potential. J Ethnopharmacology. 2002, 81, 81–100.
- Kaur, G.; Kamboj, P.; Kalia, NA. Antidiabetic and anti-hypercholesterolemic effects of aerial parts of Sida cordifolia Linn. on Streptozotocin-induced diabetic rats. Indian J Nat Prod Resour. 2011, 2, 428–434.
- Srinivasan, K.; Viswanad, B.; Asrat, L.; Kaul, LC.; Ramarao, P. Combination of high-fat diet-fed and low-dose streptozotocin-treated rat: a model for type 2 diabetes and pharmacological screening. Pharmacol Res. 2005, 52, 313–320.
Reviewer 2 Report
Nazir et al., isolated catechin from Elaeagnus umbellata Thunb. Berries and evaluated their anti-diabetic potential in SD rats. Work was well planned, executed and presented neatly. Following comments should be addressed before accepting the manuscript.
- Emphasis catechin biological properties in introduction section.
- Figure 2 A&B are same. Figure 2B shows blood glucose levels at different week. Whether Figure 2A is showing blood glucose levels at the end of treatment period? If so, it should be clearly mentioned in the figure legends.
- Symbols using for showing statistical significance (*,**,***) are same for control vs diabetes and diabetes vs CTN treated groups, which was difficult to understand. Authors can use other different symbols (#, $, ,….) for control vs diabetes and symbol (*) for diabetes vs CTN treated groups.
- In Figure 2, 3, 4 and 5, statistical significance (***) was marked in the control group (for control vs diabetes), but it should be mentioned in diabetes group.
- Why did authors used student t-test for control vs diabetes and one way ANOVA followed by Dunnett's post hoc multiple comparison test for diabetes vs CTN treated groups? Many treatment groups are used in the present study, authors should use ANOVA followed by any multiple comparison test for all the groups.
- Provide pathological score for figure 6.
- Avoid typo errors. Eg. Page 3. Line 130 “Sstructural confirmation…..”
Author Response
Reviewer 2
Nazir et al., isolated catechin from Elaeagnus umbellata Thunb. Berries and evaluated their anti-diabetic potential in SD rats. Work was well planned, executed and presented neatly. Following comments should be addressed before accepting the manuscript.
Dear Editor/ reviewer,
Thank you very much for kind review and comments concerning our manuscript. Thank you so much worthy reviewer for appreciation the manuscript data, plan of experimental study performed, and about the presentation of study. We appreciate the hard work of reviewers as they fairly pointed out errors and mistakes in our manuscript. We have tried to revise the manuscript in line with comments of the reviewers. Corrections made have been highlighted as Blue.
Please find below the point by point responses to the reviewer’s comments and suggestions.
- Emphasis catechin biological properties in introduction section.
- Answer: Worthy reviewer biological properties of catechin has been added in the introduction section in the revised manuscript accordingly.
- Figure 2 A&B are same. Figure 2B shows blood glucose levels at different week.
Whether Figure 2A is showing blood glucose levels at the end of treatment period? If so, it should be clearly mentioned in the figure legends.
- Answer: Worthy reviewer yes the Figure 2A is showing the blood glucose levels at the end of treatment period while the Figure 2B shows blood glucose levels at different week. This information has been added in the figure legends accordingly.
- Symbols using for showing statistical significance (*,**,***) are same for control vs diabetes and diabetes vs CTN treated groups, which was difficult to understand. Authors can use other different symbols (#, $, ,….) for control vs diabetes and symbol (*) for diabetes vs CTN treated groups.
- Answer: Thank you so much for your valuable comment. To reduce the confusion the statistical significance (*,**,***) the reviewer suggestion was honoured and in all figures the statistical significance (#, ##, ###) was used for control vs diabetes and (*,**,***) are used for diabetes vs CTN treated groups.
- In Figure 2, 3, 4 and 5, statistical significance (***) was marked in the control group (for control vs diabetes), but it should be mentioned in diabetes group.
- Answer: worthy reviewer thank you for the valuable suggestion. It was corrected accordingly.
- Why did authors used student t-test for control vs diabetes and one way ANOVA followed by Dunnett's post hoc multiple comparison test for diabetes vs CTN treated groups? Many treatment groups are used in the present study, authors should use ANOVA followed by any multiple comparison test for all the groups.
- Answer: Worthy reviewer the student t-test analysis compares the means (or medians) of two groups, so it is applied to only control vs diabetes. However, if the data from three or more groups, then one-way ANOVA followed by multiple comparison post tests was applied. So, one way ANOVA followed by Dunnett's post hoc multiple comparison test have been applied for diabetes vs CTN treated groups.
- Provide pathological score for figure 6.
- Answer: Worthy reviewer to calculate pathological score from these figure we requested many pathologists but they were not able to calculate the score. Being biochemist it is not possible for to calculate it. Hope the worthy reviewer will have understand our problem. The worthy reviewer may be expert and if he could provide (with request) us the necessary tools so that we would become able to do it in our future studies by ourselves.
- Avoid typo errors. Eg. Page 3. Line 130 “Sstructural confirmation…..”
- Answer: Worthy reviewer thank you very much for your sincere efforts. It was corrected accordingly.
Reviewer 3 Report
Overall the presentation of the manuscript is easy to follow. A major issue pertaining to this manuscript is the novelty of the study. A few studies have reported anti-diabetic activity of catechin in the various in-vivo models. For example:
-Catechin Treatment Ameliorates Diabetes and Its Complications in Streptozotocin-Induced Diabetic Rats
-Renoprotective effects of (+)-catechin in streptozotocin-induced diabetic rat model
-Insulin mimetic impact of Catechin isolated from Cassia fistula on the glucose oxidation and molecular mechanisms of glucose uptake on Streptozotocin-induced diabetic Wistar rats
-Catechin averts experimental diabetes mellitus-induced vascular endothelial structural and functional abnormalities
-Effects of green tea catechin on phospholipase A2 activity and antithrombus in streptozotocin diabetic rats
Line 18: Glibenclamide is the control, should be used for comparison but not to be claimed as the finding of the study.
Line 20: antiradical? should be anti-oxidant for free radical scavenging activity
Line 112 to 129: the extraction method was not valid. 500 ml of solvent unable to extract 10kg of berries. What are the extraction yield and the yield of catechin?
Line 278: how many replicates of the experiments conducted?
Line 282 and 283: The values of the % of scavenging, standard error and IC50 very similar. Please include raw data in the appendix
Line 296 and 297: The values of the % of scavenging, standard error and IC50 very similar. Please include raw data in the appendix
Author Response
Reviewer 3
Overall the presentation of the manuscript is easy to follow.
Dear Editor/ reviewer,
Thank you very much for kind review and comments concerning our manuscript. Thank you so much worthy reviewer for appreciation of the manuscript presentation. We appreciate the hard work of reviewers as they fairly pointed out errors and mistakes in our manuscript. We have tried to revise the manuscript in line with comments of the reviewers. Corrections made have been highlighted as Blue.
A major issue pertaining to this manuscript is the novelty of the study. A few studies have reported anti-diabetic activity of catechin in the various in-vivo models. For example:
-Catechin Treatment Ameliorates Diabetes and Its Complications in Streptozotocin-Induced Diabetic Rats
-Renoprotective effects of (+)-catechin in streptozotocin-induced diabetic rat model
-Insulin mimetic impact of Catechin isolated from Cassia fistula on the glucose oxidation and molecular mechanisms of glucose uptake on Streptozotocin-induced diabetic Wistar rats
-Catechin averts experimental diabetes mellitus-induced vascular endothelial structural and functional abnormalities
-Effects of green tea catechin on phospholipase A2 activity and antithrombus in streptozotocin diabetic rats
- Answer: Worthy reviewer as catechin has been isolated from many medicinal plants with beneficial health effects. However catechin isolated in this study is for the first time from the selected berries of Elaeagnus umbellata. As in Phytochemistry paper are published containing data about novel and new compounds. So we are here publishing as new compound from this plant. And as you know there is strong relationship between berries fruits and diabetes cure. Berry fruits have shown antidiabetic potential so we introduce Elaeagnus umbellata (silver berry) fruits/berry for the first time to check their antidiabetic potential. Our previous study was limited only to crude extract and their fraction which have shown both in vitro and in vivo antidiabetic potential [1]. So keeping the antidiabetic potential of this plant we plan this current study and isolate potent antidiabetic compounds from berries fruits of Elaeagnus umbellata like catechin for the first time from this plant.
Line 18: Glibenclamide is the control, should be used for comparison but not to be claimed as the finding of the study.
- Answer: Thank you worthy reviewer there may be some confusion in the context otherwise we have used it for comparison purposes. The text about its detail have been rephrased accordingly.
Line 20: antiradical? should be anti-oxidant for free radical scavenging activity
- Answer: Thank you worthy reviewer it was Corrected accordingly
Line 112 to 129: the extraction method was not valid. 500 ml of solvent unable to extract 10 kg of berries. What are the extraction yield and the yield of catechin?
- Answer: Worthy reviewer 500 mL was mistakenly added which has been removed and the extraction process has been corrected accordingly. That was used in fractionation experiments. Worthy reviewer Specified amount of crude extract was dissolved in 500 mL distilled water in a separating funnel and partitioned with different solvents starting from a low to high polarity (n-hexane, chloroform, ethyl acetate and n-butanol). The final semisolid mass obtained was solidified in open air (final mass = 750 g). The final yield of catechin obtain is 2.5 g.
Line 278: how many replicates of the experiments conducted?
- Answer: Worthy reviewer the experiment was done in three replicates. The information has been added in the revised manuscript.
Line 282 and 283: The values of the % of scavenging, standard error and IC50 very similar. Please include raw data in the appendix
- Answer: Respected reviewer Table S1 is incorporated in the revised manuscript as supplementary file to reduce the confusion related the data “In vitro antioxidant potential of CTN’’.
Line 296 and 297: The values of the % of scavenging, standard error and IC50 very similar. Please include raw data in the appendix
- Answer: Respected reviewer Table S2 is incorporated in the revised manuscript as supplementary file to reduce the confusion related the data ‘‘In vitro antidiabetic potential of CTN’’.
Round 2
Reviewer 3 Report
Minor error
line 326: the standard error/deviation is too big to be significant 65±135
Author Response
Reviewer 3:
Minor error
line 326: the standard error/deviation is too big to be significant 65±135
- Worthy reviewer, sorry it was mistake. It was 65±1.35 which have been corrected in the revised paper and supplementary material accordingly. Thank you very much for correcting the mistake.
This manuscript is a resubmission of an earlier submission. The following is a list of the peer review reports and author responses from that submission.